# KIC (ketoisocaproic acid) and leucine have divergent effects on tissue insulin signaling but not on whole-body insulin sensitivity in rats

**Gagandeep Mann, Stephen Mora, Olasunkanmi A. John Adegoke** *

School of Kinesiology and Health Science and Muscle Health Research Centre, York University, Toronto, Ontario, Canada

* oadegoke@yorku.ca

**Data Availability Statement:** All relevant data are within the manuscript and its Supporting Information files.

## Abstract

Plasma levels of branched-chain amino acids and their metabolites, the branched-chain ketoacids are increased in insulin resistance. Our previous studies showed that leucine and its metabolite KIC suppress insulin-stimulated glucose uptake in L6 myotubes along with the activation of the S6K1-IRS-1 pathway. Because other tissue and fiber types can be differentially regulated by KIC, we analyzed the effect of KIC gavage on whole-body insulin sensitivity and insulin signaling *in vivo*. We hypothesized that KIC gavage would reduce whole-body insulin sensitivity and increase S6K1-IRS-1 phosphorylation in various tissues and muscle fibers. Five-week-old male Sprague-Dawley rats were starved for 24 hours and then gavaged with 0.75ml/100g of water, leucine (22.3g/L) or KIC (30g/L) twice, ten minutes apart. They were then euthanized at different time points post-gavage (0.5-3h), and muscle, liver, and heart tissues were dissected. Other sets of gavaged animals underwent an insulin tolerance test. Phosphorylation (ph) of S6K1 (Thr389), S6 (Ser235/6) and IRS-1 (Ser612) was increased at 30 minutes post leucine gavage in skeletal muscles irrespective of fiber type. Ph-S6 (Ser235/6) was also increased in liver and heart 30 minutes after leucine gavage. KIC gavage increased ph-S6 (Ser235/6) in the liver. Neither Leucine nor KIC influenced whole-body insulin tolerance, nor ph-Akt (Ser473) in skeletal muscle and heart. BCKD-E1 α abundance was highest in the heart and liver, while ph-BCKD-E1 α (Ser293) was higher in the gastrocnemius and EDL compared to the soleus. Our data suggests that only leucine activates the S6K1-IRS-1 signaling axis in skeletal muscle, liver and heart, while KIC only does so in the liver. The effect of leucine and KIC on the S6K1-IRS-1 signaling pathway is uncoupled from whole-body insulin sensitivity. These results suggest that KIC and leucine may not induce insulin resistance, and the contributions of other tissues may regulate whole-body insulin sensitivity in response to leucine/KIC gavage.

**Funding:** Funding for this work was provided by the Natural Science and Engineering Research Council of Canada (NSERC grant# RGPIN-2021-03603) and by the Faculty of Health, York University. The funders had no role in study design, data collection and analysis, decision to publish, or preparation of the manuscript.

**Competing interests:** The authors have declared that no competing interests exist.

## Introduction

Dietary proteins, in particular branched-chain amino acids (BCAAs; leucine, isoleucine and valine), stimulate muscle protein synthesis and regulate glucose homeostasis and body weight [1]. However, elevated circulating levels of BCAAs and BCAA metabolites, branched-chain α-ketoacids (BCKAs; alpha-ketoisocaproic acid (KIC), alpha-keto-beta methylvaleric acid (KMV), and alpha-ketoisovaleric acid (KIV)) are seen in insulin-resistant states like T2DM [2–6]. This poses the question as to whether increased BCAA and BCKA levels cause insulin resistance or are a symptom of insulin resistance. Many studies have shown that BCAAs increase the activation of mammalian/mechanistic target of rapamycin complex 1 (mTORC1) and p70 ribosomal protein S6 kinase-1 (S6K1). This activation results in the inhibitory phosphorylation of serine residues of insulin receptor substrate-1 (IRS-1) (Ser636, Ser312, Ser616 in humans, and Ser632, Ser307 and Ser612 in mice) [7,8] by S6K1. This ultimately leads to the degradation of IRS-1 [9] and thus hindering downstream insulin signaling [10,11]. We have shown that leucine treatment *in vitro* (myotubes) and *in vivo* reduces insulin sensitivity, and this is seen with a consistent increase in S6K1 phosphorylation [12,13]. We have also shown that KIC, the BCKA of leucine, suppressed insulin-stimulated glucose uptake in L6 myotubes along with increases in the S6K1-IRS-1 feedback loop mechanism [12,14]. However, how the body responds to the administration of KIC and the other BCKAs would be determined by interactions amongst the tissues, not just the skeletal muscles. Muscle of different fiber types might also respond differently, as indicated by the fact that BCAA administration increases phosphorylation of mTOR (Ser2448) in the extensor digitorum longus (EDL) but not in soleus muscle [15].

BCAAs are transaminated by branched-chain aminotransferase 2 (BCAT2) predominantly in skeletal muscle, producing the BCKAs. The BCKAs are then oxidatively decarboxylated by branched-chain α-ketoacid dehydrogenase (BCKD), predominantly in the liver [16]. BCKD activity is negatively regulated by phosphorylation of its E1α-subunit on Ser 293 (BCKD-E1α Ser293) by the BCKD kinase (BDK) [17,18], and positively regulated by protein phosphatase 2Cm (pp2Cm) [19,20]. The abundance and activity of these BCAA catabolic enzymes are dysregulated in insulin resistant states [21–24].

Here, we analyzed the effect of KIC and leucine gavage on insulin tolerance and insulin signaling in various tissues and different muscle fiber types. We also assessed the relative basal abundance of enzymes involved in BCAA catabolism across different tissues and across muscle types.

## Methods

### Ethics statement

All animal experiments were approved by the York University Animal Care Committee (protocol number: #2021–01, April 2023) and were conducted in line with the guidelines of the Canadian Council on Animal Care.

### Reagents and materials

KIC (#0629), leucine (#L8912), protease (#P8430) and phosphatase (#P5726) inhibitor cocktails, o-Phthalaldehyde (OPA, #P1378) and 1,2-diamino-4,5-methylenedioxybenzene (DMB, #66807) were purchased from Sigma Aldrich (Ontario, Canada). Water, KIC and leucine were gavaged using feeding tubes (#FTP-18-75) purchased from Instech Laboratories, (Pennsylvania, United States). Insulin (Humulin R, DIN#00586714) was purchased from Eli Lilly Canada Inc. (Ontario, Canada). Glucometer (#71675) and glucose strips (#71681) were purchased

from AlphaTrak, (Ohio, United States). Antibodies against phosphorylated (ph) S6K1 (Thr389, #9234), S6 (Ser235/6, #4858), Akt (Ser473, #4060), BCKD-E1α (Ser293, #40368), IRS-1 (Ser612, #3203) and total BCKD-E1α (#90198), as well as horseradish peroxidase (HRP)-conjugated anti-rabbit (#7074) and anti-mouse (#7076) secondary antibodies were purchased from Cell Signaling Technology (Danvers, MA). Antibody against BCAT2 (#16417-1-AP) and pp2Cm (#14573-1-AP) were purchased from ProteinTech (Rosemont, IL). Antibody against BDK (#PA5-31455) was purchased from Thermo Fisher Canada (Burlington, Ontario, Canada).

## Animals

Young male Sprague–Dawley rats (150–200 g) were purchased from Charles River Laboratories Inc. (Quebec, Canada). They were acclimatized for 1 week in the animal care facility at York University while being maintained at the standard 12:12-h light–dark cycle at 22–23˚C. They had access to rat chow (product #D12450B, Research Diets, New Brunswick, NJ) and water. Animals were handled 2–3 times per week to reduce stress of handling them on the day of the experiment.

Rats were food deprived for 24 h but had access to water. They were then randomly divided into three groups. The first group was gavaged with double distilled water (ddH$_2$O) at a dose of 1.5 mL/100 g body weight in two halves, 10 minutes apart. The second group was gavaged with leucine (1 g/44.84 mL ddH$_2$O (0.170 mM), 1.5 mL/100 g body weight) in two halves, 10 minutes apart. This is equivalent to 0.33 g leucine/kg of body weight, which represents ~30% of a rat's daily leucine consumption [25]. The third group was gavaged with KIC (1 g/33.33 mL ddH$_2$O (0.197 mM), 1.5 mL/100 g body weight) in two halves, 10 minutes apart. Rats were euthanized via decapitation at different times (0.5–3 h) after gavage, and blood was taken and soleus, EDL, gastrocnemius, liver, and heart were dissected, flash-frozen in liquid nitrogen and stored at -80˚C for further analyses. To examine the effect of muscle fiber type differences in response to KIC/leucine gavage, the soleus was used for type I fibers, the EDL for type II fibers, and gastrocnemius for a mixed-fiber type muscle.

## Insulin tolerance test

Rats were food deprived for 6 h (instead of 24 h to prevent blood glucose levels dropping too low during the test) and then gavaged with either water, leucine or KIC as described above. One hour following the first gavage (this interval was chosen in order to reduce the stress of consecutive gavages/injections in a close time frame, in line with the recommendation of the institutional animal care committee), glucose readings and blood was collected through the saphenous vein. Then, an intraperitoneal insulin injection was administered (1 U/kg body weight). As done for basal glucose readings, blood samples were collected through the saphenous vein at 10–120 min post insulin administration and used to measure glucose levels.

## Tissue homogenization

Whole gastrocnemius, soleus, EDL, and sections of the liver and heart were weighed and homogenized in 7 volumes of homogenization buffer (in mM: 20 HEPES pH 7.4, 2 EGTA, 50 NaF, 100 KCl, 0.2 EDTA, 50 glycerol phosphate) supplemented with 1 mM DTT, 1 mM benzamidine, 0.5 mM sodium vanadate, and protease (10 μl/ml of homogenization buffer) and phosphatase (10 μl/ml of homogenization buffer) inhibitor cocktails were added while on ice. This homogenate was centrifuged 3 min at 1000 *g* and 4˚C. The supernatant was collected and centrifuged 30 min at 10000 g and 4˚C. The supernatants were then used for western blotting and for ultra-high-pressure liquid chromatography analyses.

## Western blotting

Proteins in homogenate were separated on 10% SDS-polyacrylamide gel electrophoresis (SDS-PAGE) followed by transfer onto polyvinylidene difluoride (PVDF) membranes (0.2 μm pore size). Incubation in primary and secondary antibodies, image acquisition, and quantification were as described [13,26].

## Amino acid analyses

We measured amino acids by HPLC as previously described [14,27,28]. Plasma samples were centrifuged for 15 min at 3000 g and 4°C. Plasma supernatants and supernatants from homogenized muscle, liver, and heart samples were diluted in a 1:2:1:8 ratio (sample: potassium phosphate buffer: 0.1 N hydrochloric acid: HPLC grade water, respectively). Diluted samples were pre-column derivatized with a 1:1 ratio of sample to OPA (29.28 mM). They were injected into a YMC-Triart C18 column (C18, 1.9 μm, 75 × 3.0 mm; YMC America, Allentown, PA, USA) fitted onto an ultra-high-pressure liquid chromatography system (Nexera X2, Shimadzu, Kyoto, Japan) that was connected to a fluorescence detector (Shimadzu, Kyoto, Japan; excitation: 340 nm; emission: 455 nm). Amino acids were eluted with a gradient solution derived from mobile phase A (20 mM potassium phosphate buffer (6.5 pH)) and mobile phase B (45% HPLC-grade acetonitrile, 40% HPLC-grade methanol and 15% HPLC-grade water) at a flow rate of 0.8 ml/min. We used a gradient of 5%–100% of mobile phase B over 21 min. Amino acid concentrations were calculated using amino acid standard curves. For muscle, liver, and heart, data were normalized to total protein concentration of the respective samples.

## KIC analyses

Protocol was adapted from Fujiwara et al [29]. Supernatants from plasma and from homogenized tissue samples were diluted in a 1:2:1:8 ratio (sample: potassium phosphate buffer: HPLC-grade water/homogenization buffer: HPLC-grade water, respectively). Diluted samples were treated with a 1:1 ratio of a DMB (1,2-diamino-4,5-methylenedioxybenzene) solution, a fluorescent agent [29]. DMB solution was prepared by adding 1.6 mg of DMB to 1.0 mL of solution that contained 4.9 mg of sodium sulfite, 70 μL of 2-mercaptethanol, and 58 μL of concentrated 12 M HCl in 0.87 mL of ddH$_2$O. Once samples were mixed with DMB, this solution was heated at 85°C for 45 minutes then cooled on ice for at least 5 min. They were injected into an Inertsil ODS-4 column (2 μM, 100 × 2.1 mm; GL Sciences, Torrance, CA, USA) fitted onto an ultra-high-pressure liquid chromatography system (Nexera X2, Shimadzu, Kyoto, Japan) that was connected to a fluorescence detector (Shimadzu, Kyoto, Japan; excitation: 367 nm; emission: 446 nm). Mobile phases were (A) HPLC-grade methanol/ddH$_2$O (30/70, v/v) and (B) HPLC-grade methanol. Gradient elution was performed as follows: 0 min 0% B, 3.33 min 0%B, 5 min 50%B, 17.34 min 50%B. The flow rate was 0.2 mL/min, and the column temperature was maintained at 40°C.

## Branched Chain Ketoacid Dehydrogenase (BCKD) assay

Protocol was adapted from White et al [24]. Frozen gastrocnemius, soleus, EDL, heart, or liver was crushed in liquid nitrogen and homogenized (Bio-Gen PRO200 Homogenizer, Connecticut, United States) in 250 μL of ice-cold buffer 1 (30 mM KPI, 3 mM EDTA, 5 mM DTT, 1 mM valine, 3% FBS, 5% Triton X-100, 1 μM leupeptin) and centrifuged at 10,000g for 10 minutes. Then, 50 μL of the supernatant was added to 300 μL of buffer 2 (50 mM HEPES, 30 mM KPI, 0.4 mM CoA, 3 mM NAD+, 5% FBS, 2 mM thiamine, 2 mM magnesium chloride and 7.8 μM $^{14}$C-labelled valine). This reaction took place in a 1.5 mL Eppendorf tube, which had a

2 M NaOH wick taped to inside surface of the lid. Each Eppendorf tube was capped and sealed tight using tape, before being placed in an incubator at 37˚C for 30 min. The radiolabeled $^{14}CO_2$ released from the BCKD reaction combined with NaOH on the wick. The wick was counted in a liquid scintillation counter to measure BCKD activity. Measured BCKD activity was corrected for the total radioactive in the reactions and to protein concentrations of the respective samples.

### Statistical analysis

Proteins for western blots were adjusted for loading using γ-tubulin values. Phosphorylated BCKD-E1α (Ser293) was also corrected to the total BCKD-E1α (Fig 7). Statistical analyses were performed using Prism 8 software (GraphPad, Massachusetts United States). Data are presented as mean ± SEM. One-way analysis of variance (ANOVA) was used. Statistically significant differences among means were determined with Tukey's post-hoc tests. Significance was determined as $p < 0.05$.

## Results

### Increase in plasma leucine and KIC from leucine and KIC gavage does not affect insulin tolerance

Plasma leucine concentrations increased 30 (p<0.0001) and 60 min (p<0.001) post leucine gavage but dropped by 120 min. KIC gavage had no effect on plasma leucine concentrations (Fig 1A). Plasma KIC levels increased 30 minutes post leucine and post KIC gavage (Fig 1B). The basal plasma level of leucine is about ~1.88X greater than the intracellular KIC level (water gavage groups: 19.5 μM vs 10.4 μM, Fig 1A and 1B). However, the fractional increase in leucine level in response to leucine gavage (~8 fold) was higher than the change in KIC level in response to KIC gavage (~6 fold). Neither leucine nor KIC gavage influenced plasma valine (S1A Fig), isoleucine (S1B Fig), or glutamate (S1C Fig) levels. In spite of the increases in their plasma levels, neither leucine nor KIC gavage affected insulin tolerance (Fig 1C and 1D), although animals that were gavaged with leucine had greater glucose AUC than those given KIC (Fig 1D, p<0.05).

### Leucine gavage, but not KIC, upregulates S6K1-IRS-1 signalling in the gastrocnemius muscle

Intracellular leucine concentrations in the gastrocnemius were increased 30 min post leucine gavage (p<0.001) but these levels dropped by 60 min. KIC had no effect on skeletal muscle leucine concentrations (Fig 2A). At the thirty minutes time point, KIC gavage but not leucine increased gastrocnemius muscle KIC concentrations (Fig 2B, p<0.01). The basal intracellular level of leucine is about ~19X the value for intracellular KIC (water gavage groups: ~0.033 μmol/μg of protein vs ~0.0017 μmol/μg of protein, Fig 2A and 2B). However, the fractional increases in leucine intracellular level in response to leucine gavage (~3.3 fold) and in KIC intracellular level in response to KIC gavage (~2.4 fold) are similar. Neither leucine nor KIC gavage had an effect on gastrocnemius valine (S2A Fig) and isoleucine (S2B Fig), but leucine gavage increased glutamate (S2C Fig) levels, consistent with the fact that glutamate is produced when leucine is transaminated by BCAT2 [30].

*In vitro*, leucine and KIC activate the S6K1-IRS-1 feedback loop [14,31]. Even though insulin tolerance was unchanged from leucine or KIC gavage (Fig 1C and 1D), S6K1 (Thr389) (Fig 2C and 2D, p<0.001), S6 (S235/6) (Fig 2C and 2E, p<0.0001) and IRS-1 (Ser612) (Fig 2C and 2F, p<0.0001) phosphorylation were increased 30 min post leucine gavage, but KIC had no

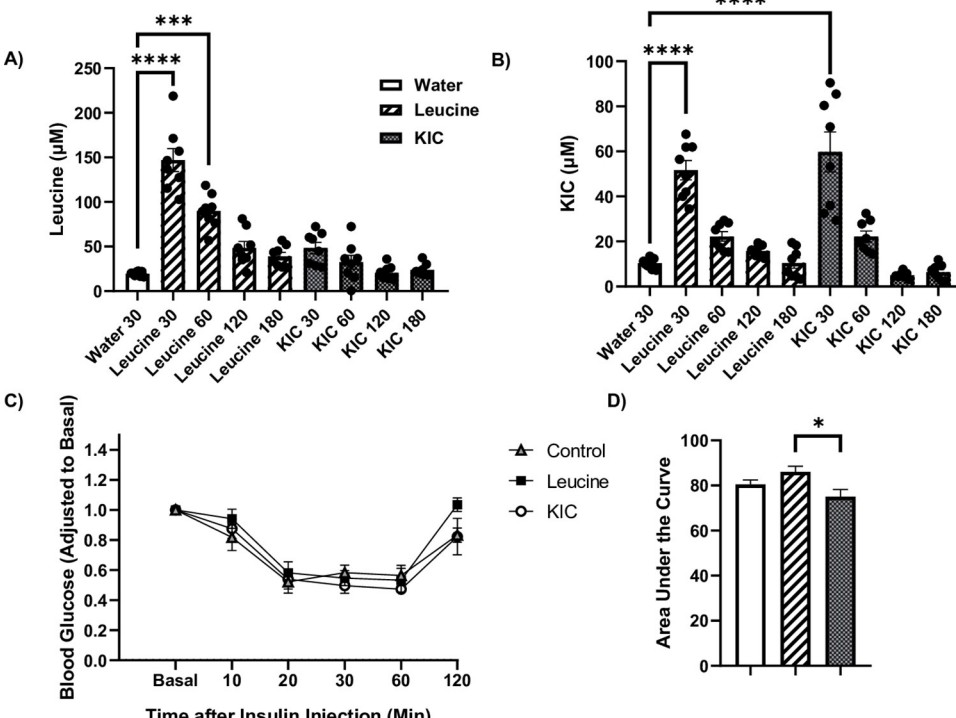

**Fig 1. Neither leucine nor KIC gavage affects whole-body insulin tolerance.** Rats were gavaged 0.75 mL/100 g body weight, twice with water, leucine (0.170 mM), or KIC (0.197 mM) 10 minutes apart. They were euthanized at different timepoints (30–180 min) after the gavage. Then HPLC was performed to measure plasma leucine (A) and KIC concentrations (B). Another group of rats was gavaged as described above. One hour after gavage, basal blood samples were taken. Animals were then administered insulin (1 U/kg of body weight). Blood samples were taken at different times to assess glucose levels (C). These blood glucose values at each timepoint were adjusted to basal values of the respective rat. Area under the curve of the blood glucose curve was analyzed (D). Data are means ± SEM, N = 8 each group; * p<0.05, *** p<0.001, **** p<0.0001.

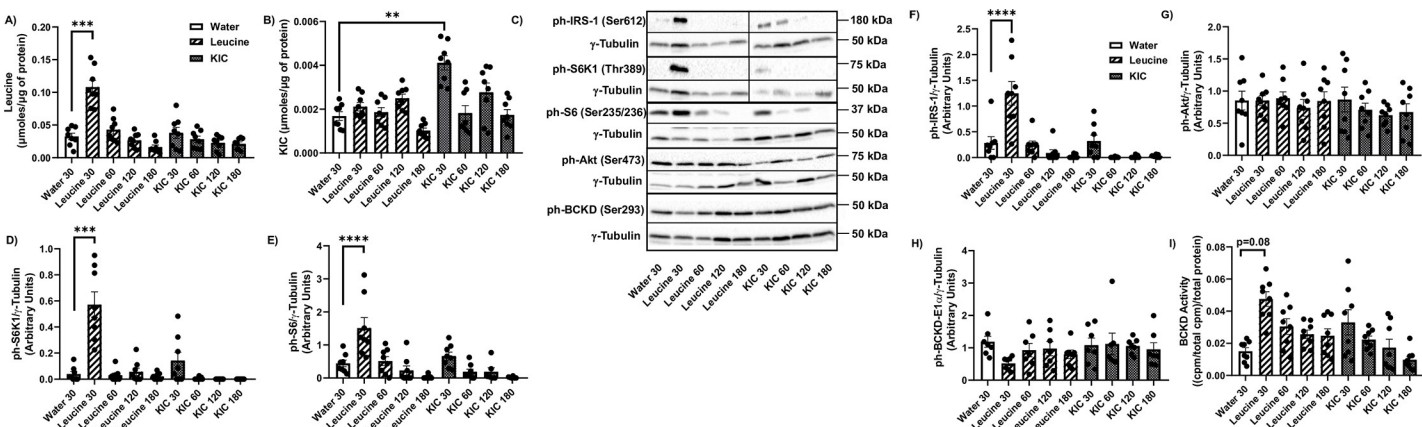

**Fig 2. Leucine, but not KIC, activates S6K1-IRS-1 signalling in the gastrocnemius muscle.** Rats were treated as explained in Fig 1. HPLC was then performed to measure intracellular leucine (A) and KIC (B) concentrations in the gastrocnemius muscle. Proteins were immunoblotted against phosphorylated (ph)-S6K1 (Thr389) (C-D), ph-S6 (Ser235/6) C, E), ph-IRS-1 (Ser612) (C, F), ph-Akt (Ser473) (C, G), and ph-BCKD-E1 α (Ser293) (C, H). BCKD activity assay was performed (I). Data are means ± SEM. N = 8 each group; ** p<0.01, *** p<0.001, **** p<0.0001.

effect on those measures (Fig 2C–2F). Neither leucine nor KIC modified Akt (Ser473) phosphorylation (Fig 2C and 2G), a marker of insulin signaling.

Next, we looked at BCKD activity, as reduced skeletal muscle BCKD activity has been linked to insulin resistance [32]. BCKD is phosphorylated when it is inhibited. Neither leucine nor KIC affected BCKD-E1α phosphorylation (Ser293) (Fig 2C and 2H), but 30 min post leucine gavage, there was trend for increased BCKD activity (Fig 2I, p = 0.08).

### Leucine, but not KIC, increases S6K1-IRS-1 signalling in the soleus muscle

BCAA supplementation increased mTOR phosphorylation in fast twitch fibers muscle such as the EDL muscle, but not in the soleus muscle [15]. Thus, we compared the effect of leucine and KIC gavage on S6K1-IRS-1 signaling in muscles of different fiber types.

In the soleus, intracellular leucine levels increased at 30 min post leucine gavage (Fig 3A, p<0.0001), and there was a trend for increased intracellular leucine level 30 min post KIC gavage (Fig 3A, p = 0.08). Also, KIC levels increased at 30 min post KIC gavage (Fig 3B, p<0,05). The basal intracellular level of leucine is about ~40X greater than the intracellular KIC level (water gavage groups: ~0.004 μmol/μg of protein vs ~0.0001 μmol/μg of protein, Fig 3A and 3B). The fractional increase in leucine level in response to leucine gavage (~6 fold) was greater than the change in KIC level in response to KIC gavage (~2 fold). Neither leucine nor KIC influenced soleus valine, isoleucine, or glutamate levels (S2A–S2C Fig).

Like the gastrocnemius, S6K1 (Thr389) (Fig 3C and 3D, p<0.001), S6 (Ser235/6) (Fig 3C and 3E, p<0.0001) and IRS-1 (Ser612) (Fig 3C–3F, p<0.0001) phosphorylation increased 30 min post leucine gavage in the soleus. KIC gavage had no significant effects on these measures (Fig 3C–3F). Neither leucine nor KIC modified Akt (Ser473) (Fig 3C and 3G) or BCKD-E1α (Ser293) (Fig 3C and 3H) phosphorylation. There was no increase in BCKD activity from leucine/KIC gavage compared to rats gavaged with water (Fig 3I), but BCKD activity was higher at 30 min compared to 120 min post leucine gavage (Fig 3I, p<0.05).

### Leucine, but not KIC, increases S6K1-IRS-1 signalling in the EDL muscle

In the EDL, leucine levels are increased 30 min (p<0.001), 60 min (p<0.01) and 120 min (p<0.05) post leucine gavage (Fig 4A). KIC gavage increased KIC levels 30 min post gavage

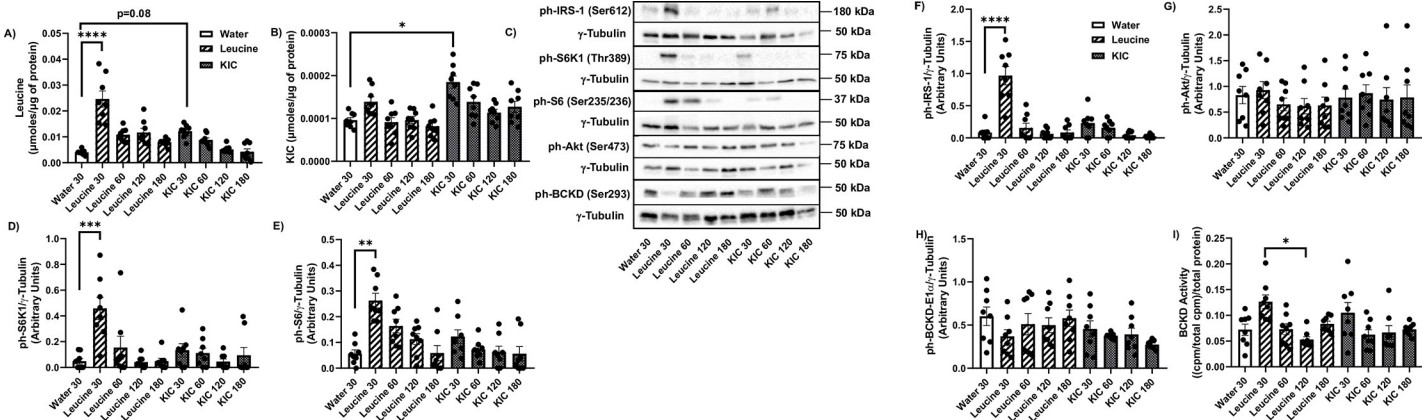

**Fig 3. Leucine, but not KIC, activates S6K1-IRS-1 signalling in the soleus muscle.** Rats were treated as explained in Fig 1. HPLC was then performed to measure intracellular leucine (A) and KIC (B) concentrations in the soleus muscle. Proteins were immunoblotted against ph-S6K1 (Thr389) (C-D), ph-S6 (Ser235/6) C, E, ph-IRS-1 (Ser612) (C, F), ph-Akt (Ser473) (C, G), and ph-BCKD-E1α (Ser293) (C, H). BCKD activity assay was performed (I). Data are means ± SEM. N = 8 each group; * p<0.05, ** p<0.01, *** p<0.001, **** p<0.0001.

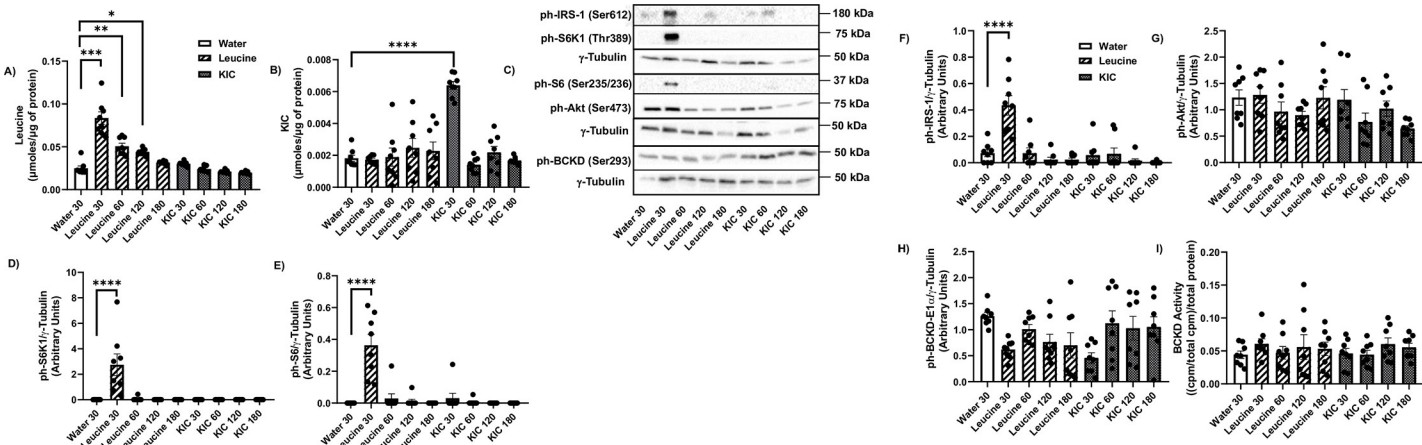

**Fig 4. Leucine, but not KIC, activates S6K1-IRS-1 signalling in the EDL muscle.** Rats were treated as explained in Fig 1. HPLC was then performed to measure intracellular leucine (A) and KIC (B) concentrations in the EDL muscle. Proteins were immunoblotted against ph-S6K1 (Thr389) (C-D), ph-S6 (Ser235/6) C, E), ph-IRS-1 (Ser612) (C, F), ph-Akt (Ser473) (C, G), and ph-BCKD-E1α (Ser293) (C, H). BCKD activity assay was performed (I). Data are means ± SEM. N = 8 each group; * p<0.05, ** p<0.01, *** p<0.001, **** p<0.0001.

(Fig 4B, p<0.0001). The basal intracellular level of leucine is about ~12.5X greater than the intracellular KIC level (water gavage groups: 0.025 μmol/μg of protein vs 0.002 μmol/μg of protein, Fig 4A and 4B). However, the fractional increase in leucine level in response to leucine gavage (~3.3 fold), and in KIC level in response to KIC gavage (~3.5 fold) was similar. Leucine but not KIC gavage tended to reduce EDL valine (S2A Fig) and significantly reduced isoleucine (S2B Fig, p<0.05) levels 60 min post leucine gavage. Leucine gavage increased glutamate levels at 60- and 120-min post gavage (S2C Fig).

Like in the gastrocnemius and soleus muscles, S6K1 (Thr389), S6 (Ser235/6), and IRS-1 (Ser612) phosphorylation was increased 30 min post leucine, but not post KIC gavage (Fig 4C–4F). Compared to what was observed for the other muscles, the responses to leucine gavage in the EDL was more pronounced and acute: there were barely any phosphorylated proteins (IRS-1, S6K1, or S6) before or at time points beyond 30 minutes after the gavage. Neither leucine nor KIC modified Akt (Ser473) (Fig 4C and 4G), BCKD-E1α (Ser293) phosphorylation (Fig 4C and 4H) or BCKD activity in the EDL (Fig 4I).

## Leucine and KIC upregulate BCKD activity and S6 phosphorylation in the liver

Liver intracellular leucine levels were increased 30 min after leucine gavage but not after KIC gavage (Fig 5A). KIC gavage increased liver KIC levels at 30 minutes post gavage (Fig 5B, p<0.05). Basal intracellular level of leucine is about ~20X greater than the intracellular KIC level (water gavage groups: 0.030 μmol/μg of protein vs 0.0015 μmol/μg of protein, Fig 5A and 5B). The fractional increase in leucine level in response to leucine gavage (~10 fold), was greater than the change in KIC level in response to KIC gavage (~2 fold). There was no effect of leucine or KIC gavage on liver valine, isoleucine, or glutamate levels (S3A–S3C Fig).

For technical reasons, we could not obtain blots for phosphorylated S6K1 in the liver and the heart, but we measured the phosphorylation of its downstream target S6. S6 (Ser235/6) phosphorylation was increased 30 min post leucine (Fig 5C and 5D, p<0.0001) and KIC gavage (Fig 5C and 5D, p<0.01), with no effect on IRS-1 (Ser612) (Fig 5C and 5E) or BCKD-E1α (Ser293) (Fig 5C and 5F) phosphorylation. Unlike in the muscles tested, there was a

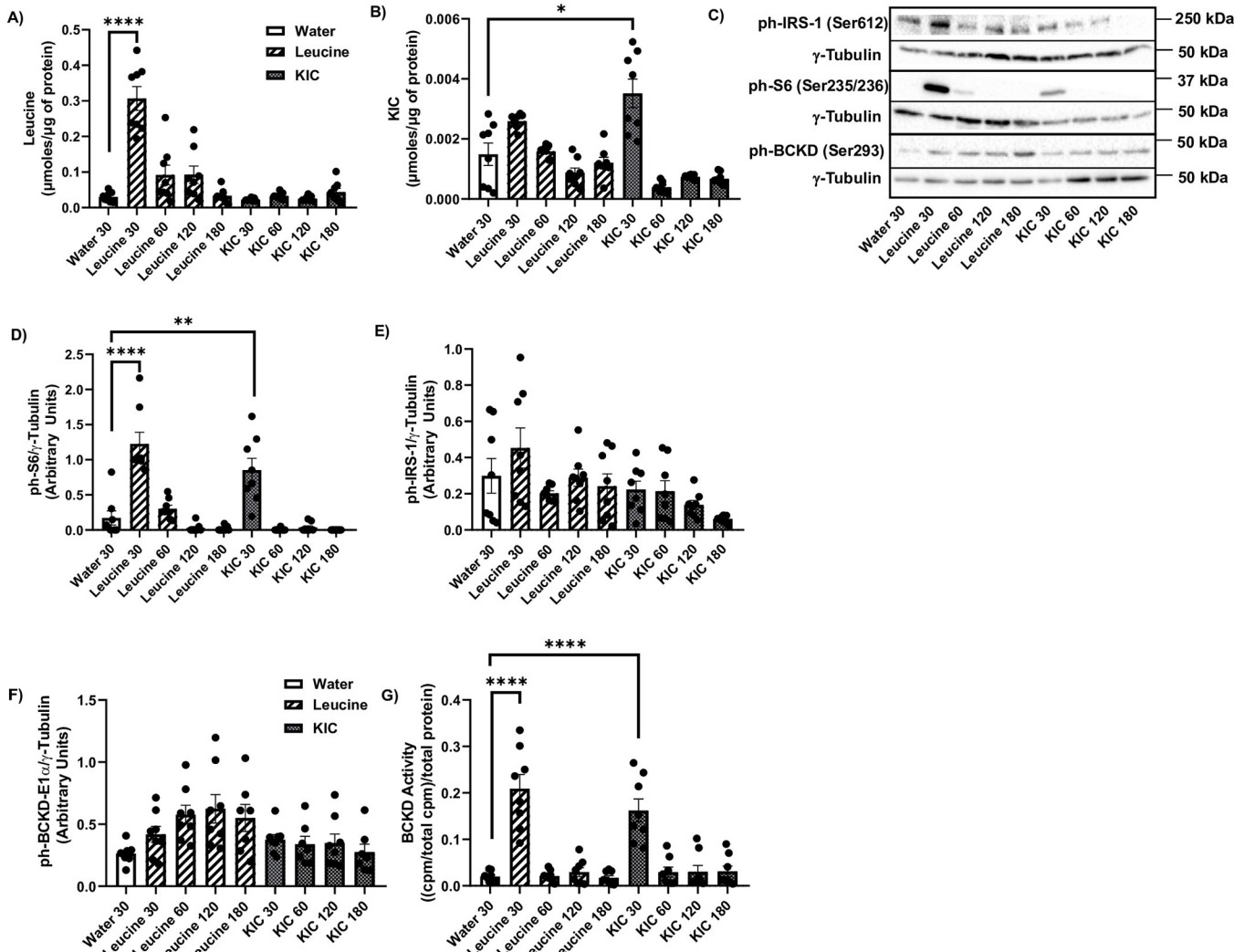

**Fig 5. Leucine and KIC increase liver BCKD activity and phosphorylation of S6.** Rats were treated as explained in Fig 1. HPLC was then performed to measure intracellular leucine (A) and KIC (B) concentrations in the liver. Liver proteins were immunoblotted against ph-S6 (Ser235/6) (C-D), ph-IRS-1 (Ser612) (C, E), and ph-BCKD-E1 α (Ser293) (C, F). BCKD activity assay was performed (G). Data are means ± SEM. N = 7–8 each group; * p<0.05, ** p <0.01, **** p<0.0001.

significant increase in liver BCKD activity 30 minutes post leucine and KIC gavage (Fig 5G, p<0.0001), consistent with the fact that the liver is the main site of BCAA oxidation.

## Leucine increases S6 phosphorylation in the heart

We next analyzed the effect of KIC gavage in the heart, as myocardial levels of BCAAs and BCKAs are also linked to cardiac insulin resistance [33]. There is a strong link between dysregulated BCAA oxidation and contractile dysfunction in heart failure [33]. The response to leucine and KIC gavage was more prolonged compared to that of any of the other muscles or the liver. Leucine levels increased 30–120 min post leucine gavage, and 30- and 60-min post KIC gavage (Fig 6A, p<0.05). Unlike what was observed for the other muscles and the liver, there was no significant effect of leucine or KIC gavage on KIC levels in the heart (Fig 6B). The basal intracellular level of leucine is about ~5X than the intracellular KIC level (water gavage groups:

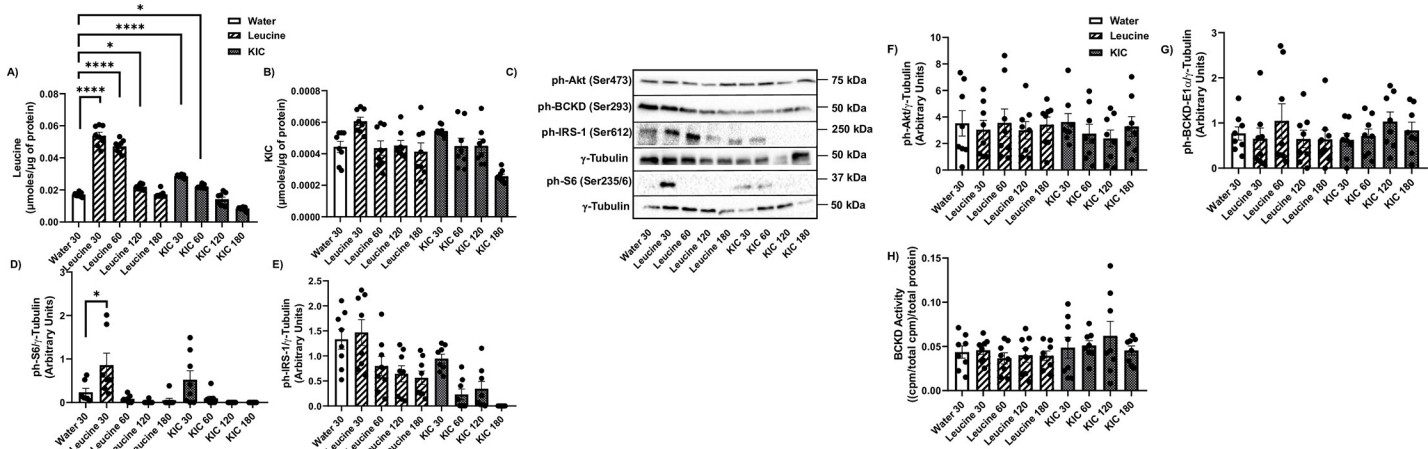

**Fig 6. Leucine gavage upregulates S6 phosphorylation in the heart.** Rats were treated as explained in Fig 1. HPLC was then performed to measure intracellular A) leucine and B) KIC concentrations in the heart. Heart proteins were immunoblotted against ph-S6 (Ser235/6) (C-D), ph-IRS-1 (Ser612) (C, E), ph-Akt (Ser473) (C, F), and ph-BCKD-E1α (Ser293) (C, G). BCKD activity assay was performed (H). Data are means ± SEM. N = 7–8 each group; * p<0.05, **** p<0.0001.

~0.002 μmol/μg of protein vs 0.0004 μmol/μg of protein, Fig 6A and 6B). Also, the fractional increase in leucine level in response to leucine gavage (~3 fold), was greater than the change in KIC level in response to KIC gavage (~1.2 fold). There was no effect of leucine or KIC on valine, isoleucine or glutamate levels (S3A–S3C Fig).

Comparatively, we note that the intracellular levels of leucine (~0.06 μmol/μg vs ~0.3 μmol/μg) and KIC (~0.0006 μmol/μg vs ~0.004 μmol/μg) are much lower in the heart compared to the liver.

In the heart, there was an increase in ph-S6 (Ser235/6) 30 min post leucine gavage (Fig 6C and 6D, p<0.05), but not KIC (Fig 6C and 6D). Like in the liver, there was no effect of KIC or leucine on IRS-1 (Ser612) (Fig 6C and 6E), Akt (Ser473) (Fig 6C and 6F) or BCKD-E1α (Ser293) (Fig 6C and 6G) phosphorylation. There was also no effect of leucine or KIC on BCKD activity in the heart (Fig 6H).

## Differences in BCAA catabolic enzyme abundance across tissues/muscle fibers

To gauge the potential relative contribution of different tissues and muscle types to BCAA catabolism, we measured the basal levels of enzymes involved. Total BCKD-E1α protein abundance was greatest in the liver and heart (Fig 7A and 7B, p<0.01). Phosphorylation of BCKD-E1α (Ser293) was the highest in the gastrocnemius and EDL (Fig 7A and 7C), with a trend for increased phosphorylated BCKD-E1α (Ser293) in the gastrocnemius compared to the soleus (p = 0.1637), liver (p = 0.1265) and heart (p = 0.1359). BCAT2 levels were not different amongst the different groups (Fig 7A and 7D). Consistent with the level of phosphorylated BCKD-E1α, BDK levels were significantly higher in the gastrocnemius compared to the other muscle fibers, and generally lower in the heart and liver compared to the skeletal muscles (Fig 7A and 7E). Finally, there were no significant differences in pp2Cm abundance although the values appear greater in the liver and heart compared to the muscles (Fig 7A and 7F).

## Discussion

Elevated levels of plasma BCAAs [4,34,35] and BCKAs [3,34,35] are a common feature of insu resistance. It is important to determine whether this is correlational or causational. Here we

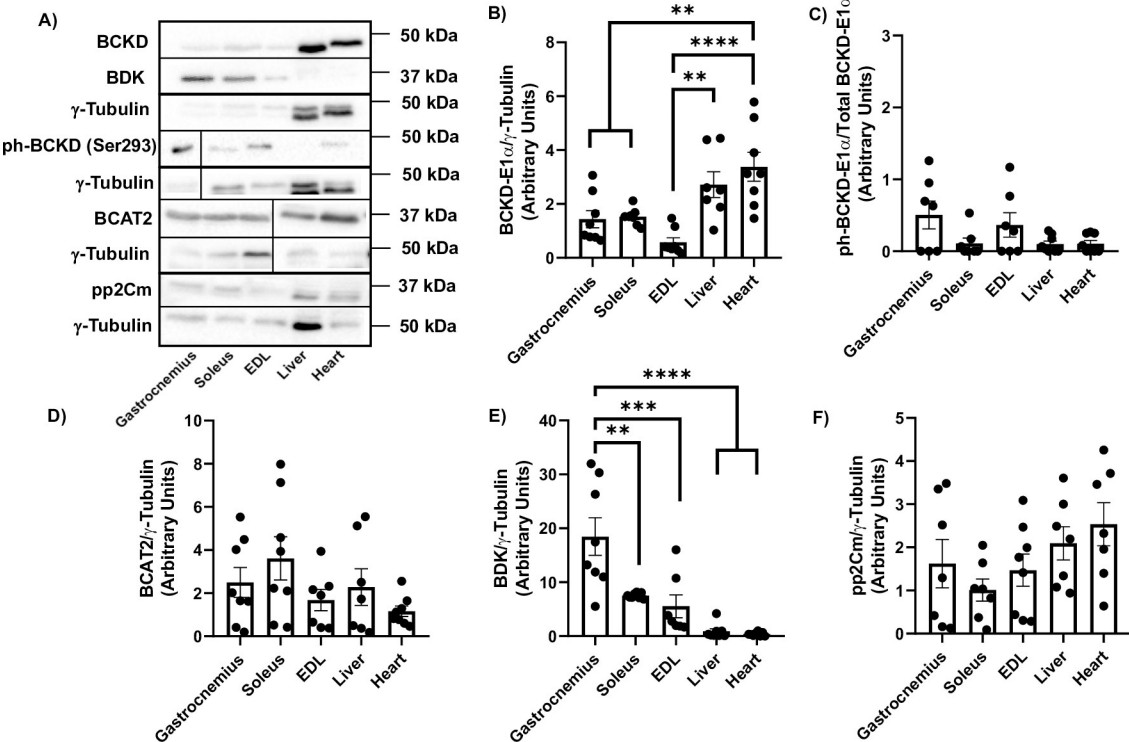

**Fig 7. Tissues/muscle fiber types have varying levels of BCAA catabolic enzyme abundance.** Gastrocnemius, soleus, EDL, liver, and heart were isolated from rats euthanized 30 min post water gavage. Proteins were immunoblotted against total BCKD-E1α (A-B), ph-BCKD-E1α (Ser293) (A, C), BCAT2 (A, D), BDK (A, E) and pp2Cm (A, F). Data are means ± SEM. N = 7–8 each group. ** $p<0.01$, *** $p<0.001$, **** $p<0.0001$.

demonstrated that the increased activation of the S6K1-IRS-1 axis in skeletal muscle (irrespective of fiber type), liver and heart in response to leucine gavage is uncoupled from whole-body insulin tolerance. We also showed that KIC gavage had no effect on S6K1-IRS-1 signalling in the heart or skeletal muscle, but increased S6 phosphorylation in the liver. Furthermore, we showed muscle- and tissue-specific differences in intracellular concentrations of leucine and KIC, and in enzymes involved in BCAA catabolism in response to leucine and/or KIC gavage. These differences may explain tissue differences that were observed in the signaling molecules that we measured.

Studies on the effect of leucine on insulin sensitivity have shown mixed outcomes, with some demonstrating improved insulin sensitivity with leucine supplementation *in vivo* [36–38], while others showed either no effect [36,39] or worsened outcome [13,40]. Also, supplementation with KIC [41,42] and leucine [43] stimulates insulin secretion *in vivo*, which could in turn reduce blood glucose levels, mitigating the effect of leucine and KIC gavage compared to the water gavage group. Interestingly, leucine and KIC at supraphysiological concentrations (2mM) stimulate glucose uptake in soleus muscle, but the effect of leucine is greater [38]. Thus, it is possible that the outcome that is obtained might be dependent on the circulating level of leucine or KIC that is attained. Circulating levels of leucine of ~120–137 μM have been reported in insulin resistant individuals [44–46], values that are close to the peak ~150 μM observed at 30 min post leucine gavage in the current study. Concentrations of KIC in the range of ~15–20 μM have been reported in ob/ob mice and diet-induced obesity [47], while plasma KIC concentrations increased from ~12 to ~24 μM in insulin resistant obese Zucker rats compared to lean counterparts [48]. Those KIC values are lower than the peak of ~60μM

observed in response to leucine or KIC gavage in the current study. The fact that we observed no significant effect of leucine/KIC gavage on whole body insulin sensitivity in this study even with circulating levels of leucine and KIC that were much higher than basal (especially for leucine, at least 3X basal at 60 min) suggests that altered leucine and KIC levels are a product of insulin resistance instead of causative. However, because the insulin tolerance test was carried out 60 min post gavage, at a time point when circulating and tissue levels of leucine and KIC were off their peak values (attained at 30 minutes), it is possible that the effects of leucine and KIC on insulin sensitivity are concentration dependent and transient. Additionally, it might also suggest that the effects of leucine and/KIC might be dependent on metabolic/health status of the individual. This is because in addition to increased circulating levels of these metabolites, elevated levels of inflammatory cytokines, which in themselves can alter insulin signaling and cause insulin resistance [49–52], are a feature of obesity/insulin resistance [53–55]. Consistent with this, in many of the tissues studied, leucine gavage exerted greater effect on mTORC1-IRS-1 signaling, a signaling axis linked to insulin resistance. This is in line with our observation of greater AUC for glucose during ITT in animals gavaged with leucine relative to those gavaged with KIC.

Previous studies have shown that mTORC1 mediates the effects of BCAAs and BCKAs on insulin resistance [34,56–59]. Activated mTORC1/S6K1 phosphorylates serine residues of IRS-1 (Ser612) [60], leading to reduced glucose transport [11,12,14]. Here we showed that leucine gavage led to increased S6K1 and IRS-1 phosphorylation 30 minutes post leucine gavage in all the muscle fibers but had no effect on phosphorylated Akt (Ser473). Another study shows that leucine protects against high fat diet induced reduction in Akt phosphorylation [61]. The lack of change in Akt phosphorylation in the various tissues in this study is also seen in another study, where one group of women consumed a whey-protein drink, while another group consumed a leucine drink. The leucine group showed an increase in mTORC1 activation in the quadricep femoris muscle, but no change in Akt activation or leg glucose uptake. On the other hand, whey-protein increases mTORC1 activation and reduces leg glucose uptake with no change in Akt activation [62]. These data are consistent with our report and suggest that the effect of leucine/protein ingestion on glucose uptake/insulin sensitivity is not due to inhibition of Akt.

Yamanashi et al. showed that BCAAs increase mTOR phosphorylation in the EDL, but not the soleus [15]. The soleus is a more oxidative muscle, which has more mitochondria than fast twitch muscles like the EDL [63]. We could not directly compare BCKD activity because the assays for the different muscle types were done on different days. However, in response to leucine gavage, the fractional increase in BCKD activity in the soleus was ~43% versus a ~22% increase in the EDL. Intracellular BCAA and KIC levels are higher in the EDL and gastrocnemius than the soleus in the basal group (Figs 2A, 2B, 4A and 4B compared to Figs 3A and 4B; also see S2 Fig). These differences in BCAA/KIC levels could be due to lower phosphorylated BCKD-E1α (Ser293) and higher BCKD activity in the soleus (Fig 7C) and thus greater BCAA catabolism. This is also consistent with a much greater leucine to KIC ratio in the soleus (40:1) compared to the gastrocnemius (19: 1) or the EDL (12.5:1). Whether these differences are due solely to muscle fiber-type specific differences in BCKD activity, and/ or differences in the use of leucine for protein synthesis or differences in transport of KIC and or leucine require further experiments.

A question arising from muscle-type differences in BCKD activity/BCAA oxidation and KIC/leucine intracellular levels is whether these differences have any relevance vis-à-vis insulin sensitivity in muscle of different fiber types. While it is known that type II fiber muscles such as the EDL tend to be more plastic in response to atrophy-inducing insults [64], it is not clear whether there is fiber-type differences in insulin sensitivity. There were no striking differences

amongst the muscles in mTORC1/IRS-1 axis in response to leucine gavage. To the extent that this axis regulates insulin signaling in muscle, one would predict similar responses amongst the different muscles in insulin-stimulated glucose uptake in response to leucine gavage. Type II muscle fibers have greater insulin-stimulated and contraction-stimulated glucose uptake than type I fibers due to increased GLUT4 protein content in rats [65]. Also, in lean and obese humans, phosphorylation of Akt (Thr308) is higher in response to insulin in type II fibers compared to type I fibers, but this did not translate into significant differences in GLUT4 protein content [66].

In our previous *in vitro* studies, we showed that KIC is converted back to leucine to elicit its effects on glucose transport and insulin signaling [14,31]. Here we show that KIC gavage does not result in a significant increase in leucine levels in the skeletal muscles, although there was a trend for higher leucine levels in the soleus (p = 0.08). The attenuated effects of KIC gavage, relative to leucine gavage, on the mTORC1-IRS-1 signaling in tissues is likely related to body handling of these metabolites. Because these were oral gavages, that the fractional increases in liver and circulating level of leucine in response to leucine gavage are much higher than the fractional increase in KIC in response to KIC gavage is consistent with the fact that the liver (and intestinal mucosa) is not a main site of BCAA transamination [67,68] but the main site of BCKD activity [67–69]. This reasoning is consistent with the observation that it was only in the liver that KIC gavage led to increased mTORC1 signaling. However, in the skeletal muscles, especially the gastrocnemius and EDL, fractional increases were either the same or higher for KIC compared to leucine gavage, reflecting the fact that the liver, as compared to the muscle, is the main site of KIC oxidation [67–69]. This is supported by the fact the liver had higher levels of BCKD-E1α and pp2Cm but lower levels of BDK and phosphorylated BCKD-E1α (Ser293) (Fig 7) compared to muscles of diverse fiber types. One implication of these findings is that to study a direct effect of these substrates on insulin signaling, oral leucine gavage as done here is the best approach since the liver does not extract much of the BCAA that is administered. For KIC however, a better approach would be to deliver KIC intravenously in order to reduce its first pass extraction by the liver. While this approach would be scientifically logical, it imposes a limitation: the preferred way of delivering nutrients is orally.

As discussed above, changes in BCAA metabolism and its levels can affect heart function and cardiac insulin sensitivity [33]. One study demonstrated major increases in serine phosphorylation of IRS-1 in the hearts of hypertensive rats compared to controls [70]. We observed an increase in S6 (Ser235/6) phosphorylation in the heart. This suggests increased mTORC1/S6K1 activity, but this does not translate to increases in phosphorylation of IRS-1 (Ser612). The lack of change in IRS-1 (Ser612) in the heart could be due to the fact the muscles had a greater degree of S6 phosphorylation (reflective of S6K1 activity) in response to leucine gavage (p<0.0001) compared to the heart (p<0.05).

Dietary restriction of BCAAs improves insulin sensitivity in rodent models [32,71–73]. As discussed above, whey ingestion reduces leg glucose uptake while leucine does not [62]. This suggests that perhaps valine and isoleucine could be the specific culprits in BCAA-induced insulin resistance. Valine supplementation worsens the obesogenic effects and insulin sensitivity of a high fat diet while leucine protects against these [61]. This correlated with increased valine-induced BCKD inhibition [61], while in this study we showed a trend for increased BCKD activation 30 min post leucine gavage in the gastrocnemius and soleus muscles and the liver. In another study, isoleucine restriction improves insulin sensitivity in normal chow fed mice [74]. Reintroducing isoleucine attenuates the metabolic improvements of a low amino acid diet independent of S6K1-IRS-1 signalling pathway [74]. Additionally, isoleucine supplementation in mice fed a high fat diet worsens insulin resistance, along with increased intramuscular lipid droplet accumulation and mitochondrial dysfunction [75]. These findings

suggest that the adverse effects of BCAAs may be mediated by isoleucine and valine but as mentioned before, more studies using different routes of delivery (intravenous vs oral) and in animals in different metabolic/health states are needed to clarify this point.

One limitation of this study is that we did not measure IRS-2 in the liver. To our knowledge, no study has measured the effect of leucine on S6K1-IRS-2 signalling in the liver. Although one study showed that mTORC1 is required for insulin stimulation of serine phosphorylation of IRS-2 in Fao cells (hepatic carcinoma cells), S6K1 is not required for this effect. Furthermore, this phosphorylation had no effect on the downstream PI3K/Akt pathway [76]. Another limitation is that we could not obtain signals for ph-S6K1 (Thr389) in the liver and heart. There was a signal for total S6K1 in both tissues, suggesting that the effect of leucine or KIC gavage on S6K1 phosphorylation is more transient in these tissues than in skeletal muscles. Additionally, we performed the insulin tolerance test 60 min after gavage instead of at 30 min (when circulating and tissue leucine and KIC levels peaked). This decision was made based on the recommendation of the institutional animal care committee in order to reduce the stress of consecutive gavages/injections in a close time frame. By so doing however, we might have missed a critical time point because it is possible that the effect of the gavage on insulin sensitivity is concentration dependent and transient. Finally, we had difficulty in obtaining signals for indicators of insulin signaling in adipose tissues. This tissue is a contributor to BCAA catabolism [77]. Despite this, since adipose tissue BCKD activity is low [68], it is likely that KIC is not activating the S6K1-IRS-1 pathway in adipose tissue. Also, testing the effect of an acute gavage of leucine/KIC is a limitation, as people typically consume BCAA daily, in a chronic manner. However, it is important to elucidate how leucine/KIC gavage acutely affects specific mechanisms related to health because of the increasing prevalence of the use of single amino acids for anabolic effects [78,79] and overall health [80,81]. Future long-term studies are warranted to see if long term treatment of KIC differs in its effect from an acute gavage.

In conclusion, while leucine or KIC gavage increased circulating and tissue levels of the gavaged metabolites, and leucine gavage induced mTORC1/IRS-1 signaling in muscle of diverse fiber types and in liver, neither leucine nor KIC gavage had a significant effect on whole body insulin sensitivity. There were muscle type-specific and tissue-specific differences in intracellular leucine and KIC levels and in the effect of leucine and/or KIC gavage on BCKD activity. However, these differences are likely evened out by contributions of diverse tissues in the body so that in sum, there was no effect of KIC or leucine on whole body insulin tolerance. Recent studies point to differences in each of the BCAA in contributing to insulin resistance [61,74]. Our data support the notion that while elevated circulating BCAA is a feature of insulin resistance, leucine (or KIC) does not induce insulin resistance, at least in healthy animals and within the limits of this study. However, more appropriately designed future studies are warranted to clarify this subject, especially because of the contribution of insulin resistance to major human chronic diseases.

## Supporting information

**S1 Fig. Neither leucine nor KIC gavage affects plasma valine, isoleucine, and glutamate.** Rats were gavaged 0.75 mL/100 g body weight, twice with water, leucine (0.170 mM), or KIC (0.197 mM) 10 minutes apart. They were euthanized at different timepoints (30–180 min). HPLC was performed to measure valine (A), isoleucine (B), and glutamate (C) concentrations in plasma. Data are means ± SEM. N = 8 each group.
(TIF)

**S2 Fig. Leucine gavage, but not KIC, increases glutamate levels in the gastrocnemius and EDL muscle, and decreases isoleucine levels in the EDL muscle.** Rats were treated as

explained in S1 Fig. HPLC was then performed to measure intracellular valine (A), isoleucine (B), and glutamate (C) concentrations in the gastrocnemius, soleus and EDL muscles. Data are means ± SEM. N = 7–8 each group; * p<0.05, ** p < 0.01, *** p < 0.001.
(TIF)

**S3 Fig. Neither Leucine nor KIC gavage affects valine, isoleucine, or glutamate levels in the liver or heart.** Rats were treated as explained in S1 Fig. HPLC was then performed to measure intracellular valine (A), isoleucine (B), and glutamate (C) concentrations in the liver and heart. Data are means ± SEM. N = 7–8 each group.
(TIF)

**S1 File.**
(PDF)

# Acknowledgments

GM and OAJA conceived and designed the experiments. GM and SM performed the experiments. GM drafted the initial version of the manuscript. OAJA reviewed and edited the manuscript. All authors approved the final version of the manuscript.

# Author Contributions

**Conceptualization:** Gagandeep Mann, Olasunkanmi A. John Adegoke.

**Funding acquisition:** Olasunkanmi A. John Adegoke.

**Investigation:** Gagandeep Mann, Stephen Mora.

**Methodology:** Gagandeep Mann.

**Project administration:** Gagandeep Mann, Olasunkanmi A. John Adegoke.

**Resources:** Olasunkanmi A. John Adegoke.

**Supervision:** Olasunkanmi A. John Adegoke.

**Visualization:** Gagandeep Mann.

**Writing – original draft:** Gagandeep Mann.

**Writing – review & editing:** Gagandeep Mann, Stephen Mora, Olasunkanmi A. John Adegoke.

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
