## [Decision Letter · Decision Letter 0]

3 May 2024

PONE-D-24-07286KIC (ketoisocaproic acid) and leucine have divergent effects on tissue insulin signaling but not whole-body insulin sensitivity in ratsPLOS ONE

Dear Dr. Adegoke,

Thank you for submitting your manuscript to PLOS ONE. After careful consideration, we feel that it has merit but does not fully meet PLOS ONE’s publication criteria as it currently stands. Therefore, we invite you to submit a revised version of the manuscript that addresses the points raised during the review process.

We look forward to receiving your revised manuscript.

Kind regards,

Md. Wasim Khan, Ph.D.

Academic Editor

PLOS ONE

Journal Requirements:

3. Please expand the acronym “NSERC” (as indicated in your financial disclosure) so that it states the name of your funders in full.

4. Thank you for stating the following financial disclosure: "NSERC"

**Additional Editor Comments:**

Dear Dr Adegoke

The reviewers have submitted their critiques. As you will see, Rev 2 has proposed that the manuscript be rejected in its current form. However, although I agree with all the comments of both Revs 1 and 2, I feel that if given the opportunity, this manuscript could be improved. Particularly, the comment in Line 373 has caught my attention - S6K immunoblots were not possible due to technical difficulties. There are hundreds of published articles showing S6K immunoblots, and I do not think this can be a challenge, especially given the premise of this manuscript. I invite you to revise the manuscript according to the comments.

Reviewers' comments:

Reviewer's Responses to Questions

**Comments to the Author**

1. Is the manuscript technically sound, and do the data support the conclusions?

Reviewer #1: Yes

Reviewer #2: No

2. Has the statistical analysis been performed appropriately and rigorously? 

Reviewer #1: Yes

Reviewer #2: Yes

3. Have the authors made all data underlying the findings in their manuscript fully available?

Reviewer #1: Yes

Reviewer #2: Yes

4. Is the manuscript presented in an intelligible fashion and written in standard English?

Reviewer #1: Yes

Reviewer #2: Yes

5. Review Comments to the Author

Reviewer #1: Very nice piece of work, well written and communicated. However, why didn't the authors look at PI3K and GLUT4 to confirm the insulin signaling pathway?. Also, to check if there is any interaction with PTEN since the author was looking at the activation of IRS-1,? There was no mention of the number, ethics clearance, month, or year of approval. Why did the author use HPLC to detect BCAAs and not conventional UV detectors or capillary electrophoresis (CE) coupled with conventional UV detectors?

Reviewer #2: This is a well written paper with some well thought out work, but there are few flaws that prevent it from receiving an acceptance. 1) The authors clearly demonstrate that leucine and KIC levels are elevated at 30 minutes and significantly decrease in most tissues by 60 minutes following gavage, yet the insulin tolerance tests performed occurred at 60 minutes. The authors argue based on these tolerance tests that there is no insulin resistance caused by leucine and KIC, yet their time point is clearly after levels have returned to normal physiological levels and no longer match levels cited in their discussion as being pathological. 2) The signaling pathway for IRS phosphorylation and regulation in the liver is incomplete due to the omission of IRS-2 phosphorylation status. While IRS-2 does not play a pronounced role in muscle insulin sensitivity and corresponding glucose tolerance, it plays a crucial role in the liver. 3) The statement in line 373-374 regarding the missing S6K1 and proxy evidence of S6 phosphorylation at S235/236 is not adequate, as this site has been shown to be phosphorylated by multiple other kinases. This figure needs the addition for S6K1 for any claim that the observed S6 phosphorylation is driven via AKT-mTOR1-S6K1 activity.

6. PLOS authors have the option to publish the peer review history of their article (what does this mean?). If published, this will include your full peer review and any attached files.

Reviewer #1: **Yes: **Patrick Nwabueze Okechukwu

Reviewer #2: No

---

## [Author Response · Author response to Decision Letter 0]

11 Jun 2024

We thank the associate editor and the reviewers for carefully reading our manuscript and the detailed comments submitted. We have thoroughly and substantially reviewed the manuscripts to reflect the concerns raised. In the responses below, references to page and line numbers are for the clean copy of the revised manuscript.

Associate Editor:

Authors’ Response:

We thank the associate editor for this comment and have ensured the manuscript is correctly formatted. 

Authors’ Response:

We thank the associate editor for this comment. We have indicated on page 7 line 168 that we euthanized the rats via decapitation. On page 7 line 159-60 we had already indicated that rats were handled 2-3 times weekly to reduce stress on the day of the gavage. We also included on page 8 lines 178-180, that we waited 1 hour between the first gavage and the insulin injection to reduce the stress on the animals.

3. Please expand the acronym “NSERC” (as indicated in your financial disclosure) so that it states the name of your funders in full.

Authors’ Response:

Thank you: we have made these changes in the cover letter.

4. Thank you for stating the following financial disclosure: "NSERC"

Authors’ Response:

Thank you: we have made these changes in the cover letter.

5. PLOS ONE now requires that authors provide the original uncropped and unadjusted images underlying all blot or gel results reported in a submission’s figures or Supporting Information files. 

Authors’ Response:

Thank you for this comment: we have included the uncropped and unadjusted images underlying all blot results in the supporting information files.

6) The reviewers have submitted their critiques. As you will see, Rev 2 has proposed that the manuscript be rejected in its current form. However, although I agree with all the comments of both Revs 1 and 2, I feel that if given the opportunity, this manuscript could be improved. Particularly, the comment in Line 373 has caught my attention - S6K immunoblots were not possible due to technical difficulties. There are hundreds of published articles showing S6K immunoblots, and I do not think this can be a challenge, especially given the premise of this manuscript. I invite you to revise the manuscript according to the comments.

Authors’ Response:

We thank the associate editor for this comment: 

As can be seen in the original manuscript, we obtained phosphorylated S6K1 and showed that it responded to leucine gavage in ALL the three muscles that we studied (gastrocnemius, soleus and EDL) (Fig 2C-D, Fig 3C-D, and Fig 4C-D). So, the issue was not that we were unable to obtain phosphorylated S6K1 signal per se. Rather, our challenge was with obtaining phosphorylated S6K1 signal in liver and heart samples. Hence our resolve to use the phosphorylated S6 as a surrogate for phosphorylated S6K1 (again, only for liver and heart samples). We acknowledge the fact that it would be good to have phosphorylated S6K1 data for liver and heart, and we attempted to do so but were unsuccessful. But even if we had succeeded, given the fact that in liver and heart samples, IRS-1 serine phosphorylation (which would result from the action of phosphorylated (activated) S6K1) is NOT altered (Fig 5C, E, and Fig 6C, E), nor was there treatment effect on AKT (Fig 5C, F and 6C, F), it is our opinion that any treatment effect on S6K1 phosphorylation would be inconsequential to the conclusion of the manuscript. Nevertheless, and to further address this, we prepared fresh heart and liver samples, loaded higher amounts of proteins per well than we did before, and obtained new antibodies to repeat the western blot. Unfortunately, there was again no signal for phosphorylated S6K1 in either heart or liver samples. Because we loaded control samples provided by the antibody manufacturer and successfully detected signals for phosphorylated S6K1 in those control samples, we believe the inability to detect phosphorylated S6K1 signal in those samples was because of the transient nature of S6K1 phosphorylation in response to leucine/KIC gavage in heart and liver as compared to muscle samples (please see the blots from these new analyses in the supporting information; the blots are also pasted below). Please note that unlike the signals for phosphorylated S6K1, we robustly detected total S6K1 in these samples (please see supporting information; the blots are also pasted below). 

Reviewer #1: 

Very nice piece of work, well written and communicated. However, why didn't the authors look at PI3K and GLUT4 to confirm the insulin signaling pathway?. 

Authors’ Response:

We thank the reviewer for these comments. Akt is downstream of PI3K, and Akt phosphorylation is a strong indicator of insulin signaling (Diabetes Metab Syndr Obes. 2014; 7: 55–64) as it is involved in GLUT4 translocation (Molecular Endocrinology, Volume 19, Issue 4, 1 April 2005, Pages 1067–1077). Akt was unaltered in response to KIC or leucine gavage, which is consistent with other studies that showed an increase in the activation of the mTORC1-S6K1-IRS-1 axis but without an effect on Akt in response to leucine/protein ingestion (Diabetes. 2015 May;64(5):1555-63). Given the absence of treatment effect on Akt, we do not think measuring PI3K would add much to the manuscript. 

The half-life of GLUT4 protein is 8-10 hours (Biochem Biophys Rep. 2015 Jul; 2: 45–49), longer than our treatment times. GLUT4 translocation too can be affected by treatments. However, since this translocation is regulated by AKT (World J Biol Chem. 2020 Nov 27; 11(3): 76–98), but Akt phosphorylation is unchanged in every tissue reported in this study, we think it unlikely that GLUT4 translocation would be affected under the conditions studied. One study shows that supraphysiological levels of leucine (10 mM) can reduce GLUT4 translocation in cardiomyocytes. However, this study used 6-7x greater amount of leucine, and they saw a consistent reduction in Akt phosphorylation with reduced GLUT4 (Am J Physiol Heart Circ Physiol 313: H432–H445, 2017). Thus, the lack of effect on phosphorylated Akt in our study suggests that GLUT4 translocation was not changed. GLUT4 translocation can also occur in response to exercise, but this was not studied this manuscript. 

Also, to check if there is any interaction with PTEN since the author was looking at the activation of IRS-1?

Authors’ Response:

Thank you for this comment. The half-life of PTEN is 7-8 hours (Cold Spring Harb Perspect Med. 2020 May; 10(5): a036079), so we think it unlikely that 0.5-3 h of leucine/KIC gavage would affect PTEN abundance. Also, PTEN is downstream of IRS-1 and is involved in negatively regulating PI3K. If PTEN protein level/interaction were changed, this might result in changes in the phosphorylation of Akt, which we did not see. Consistent with the reviewer’s comment, there is evidence that PTEN can dephosphorylate tyrosine residues of IRS-1 (Nat Struct Mol Biol. 2014 Jun; 21(6): 522–527). However, although a few studies examined the effects of nutrients on PTEN (Discov Oncol. 2023 Feb 23;14(1):25.doi: 10.1007/s12672-023-00634-1; Nutr. 2021 Sep 4;151(9):2636-2645.doi: 10.1093/jn/nxab190), much of what is known about the phosphatase relates to its regulation by growth factors. In future, it would be interesting to study whether amino acids regulate its interactions and whether such a regulation is functionally consequential.

 There was no mention of the number, ethics clearance, month, or year of approval. 

Authors’ Response:

Thank you. On page 6 lines 136-138 we included an ethics statement, the animal use protocol number, month and year of approval. 

Why did the author use HPLC to detect BCAAs and not conventional UV detectors or capillary electrophoresis (CE) coupled with conventional UV detectors?

Authors’ Response:

We thank the reviewer for this comment. Although amino acids may be measured using different systems, liquid chromatography (HPLC) is one of the common systems and has been our preferred approach (Am J Physiol Cell Physiol. 2024 Mar 1;326(3):C866-C879, Physiol Rep. 2024 Apr; 12(8): e16003, Physiol Rep. 2021 Jan; 9(1): e14673). We have now included this in the methods section on page 9 lines 202. 

Reviewer #2: 

1) The authors clearly demonstrate that leucine and KIC levels are elevated at 30 minutes and significantly decrease in most tissues by 60 minutes following gavage, yet the insulin tolerance tests performed occurred at 60 minutes. The authors argue based on these tolerance tests that there is no insulin resistance caused by leucine and KIC, yet their time point is clearly after levels have returned to normal physiological levels and no longer match levels cited in their discussion as being pathological. 

Authors’ Response:

We thank the reviewer for taking the time to read our manuscript and for the reviewer’s comments. Regarding when the ITT was done, firstly, the institutional animal care committee had us wait an hour after the first gavage before doing the ITT in order to reduce the amount of stress on the animals. Secondly, although leucine levels dropped after 60 min in most tissues, the level was still quite high in plasma (~100�M), a level not far from those seen in insulin resistance as discussed on pages 21-22, lines 485-490. Even if there was an insulin resistance at an earlier time point that we missed, this would still suggest that, at best, such an effect was very transient and might be inconsequential.

2) The signaling pathway for IRS phosphorylation and regulation in the liver is incomplete due to the omission of IRS-2 phosphorylation status. While IRS-2 does not play a pronounced role in muscle insulin sensitivity and corresponding glucose tolerance, it plays a crucial role in the liver. 

Authors’ Response:

We thank the reviewer for this comment. We agree that IRS-2 is crucial in the liver, to our knowledge IRS-2 is not phosphorylated by S6K1 in the liver in response to leucine or KIC (or any BCAA/BCKAs), the nutrients that we studied. However, S6K1 liver-specific knockdown resulted in increased total IRS-2 protein, suggesting that S6K1 could impact IRS-2 (J Biol Chem. 2012 May 25; 287(22): 18769–18780), thus we have included the omission of IRS-2 as a limitation in our discussion on page 26 line 588. Others have looked at how S6K1 phosphorylates IRS-1 serine residues as reviewed (Nutrients. 2016 Jul; 8(7): 405), and specifically in the liver (Proc Natl Acad Sci USA. 2007 Aug 28;104(35):14056-61). Although one study shows that mTORC1 is required for insulin stimulation of serine phosphorylation of IRS-2 in Fao cells (hepatic carcinoma cells), S6K1 is not required for this effect. Furthermore, this phosphorylation had no effect on the downstream PI3K/Akt pathway (Am J Physiol Endocrinol Metab. 2011 May;300(5):E824-36). We have included this in the discussion section on page 26, lines 588-592.

3) The statement in line 373-374 regarding the missing S6K1 and proxy evidence of S6 phosphorylation at S235/236 is not adequate, as this site has been shown to be phosphorylated by multiple other kinases. This figure needs the addition for S6K1 for any claim that the observed S6 phosphorylation is driven via AKT-mTOR1-S6K1 activity.

Authors’ Response:

We thank the reviewer for this comment.

As can be seen in the original manuscript, we obtained phosphorylated S6K1 and showed that it responded to leucine gavage in ALL the three muscles that we studied (gastrocnemius, soleus and EDL) (Fig 2C-D, Fig 3C-D, and Fig 4C-D). So, the issue was not that we were unable to obtain phosphorylated S6K1 signal per se. Rather, our challenge was with obtaining phosphorylated S6K1 signal in liver and heart samples. Hence our resolve to use the phosphorylated S6 as a surrogate for phosphorylated S6K1 (again, only for liver and heart samples). We acknowledge the fact that it would be good to have phosphorylated S6K1 data for liver and heart, and we attempted to do so but were unsuccessful. But even if we had succeeded, given the fact that in liver and heart samples, IRS-1 serine phosphorylation (which would result from the action of phosphorylated (activated) S6K1) is NOT altered (Fig 5C, E, and Fig 6C, E), nor was there treatment effect on AKT (Fig 5C, F and 6C, F), it is our opinion that any treatment effect on S6K1 phosphorylation would be inconsequential to the conclusion of the manuscript. Nevertheless, and to further address this, we prepared fresh heart and liver samples, loaded higher amounts of proteins per well than we did before, and obtained new antibodies to repeat the western blot. Unfortunately, there was again no signal for phosphorylated S6K1 in either heart or liver samples. Because we loaded control samples provided by the antibody manufacturer and successfully detected signals for phosphorylated S6K1 in those control samples, we believe the inability to detect phosphorylated S6K1 signal in those samples was because of the transient nature of S6K1 phosphorylation in response to leucine/KIC gavage in heart and liver as compared to muscle samples (please see the blots from these new analyses in the supporting information; the blots are also pasted below). Please note that unlike the signals for phosphorylated S6K1, we robustly detected total S6K1 in these samples (please see supporting information; the blots are also pasted below).

---

## [Decision Letter · Decision Letter 1]

16 Jul 2024

PONE-D-24-07286R1KIC (ketoisocaproic acid) and leucine have divergent effects on tissue insulin signaling but not on whole-body insulin sensitivity in ratsPLOS ONE

Dear Dr. Adegoke,

Thank you for submitting your manuscript to PLOS ONE. After careful consideration, we feel that it has merit but does not fully meet PLOS ONE’s publication criteria as it currently stands. Therefore, we invite you to submit a revised version of the manuscript that addresses the points raised during the review process.

 The reviewer has raised an important issue that needs to be addressed. In particular, the timing of insulin tolerance tests relative to detection of KIC and Leucine levels has not been adequately addressed in the revised manuscript. We are therefore requesting the authors to address this and the other comments from the reviewer in the discussion and study limitations sections.

We look forward to receiving your revised manuscript.

Kind regards,

Owen Ngalamika

Academic Editor

PLOS ONE

Reviewers' comments:

Reviewer's Responses to Questions

**Comments to the Author**

1. If the authors have adequately addressed your comments raised in a previous round of review and you feel that this manuscript is now acceptable for publication, you may indicate that here to bypass the “Comments to the Author” section, enter your conflict of interest statement in the “Confidential to Editor” section, and submit your "Accept" recommendation.

Reviewer #2: (No Response)

2. Is the manuscript technically sound, and do the data support the conclusions?

Reviewer #2: Partly

3. Has the statistical analysis been performed appropriately and rigorously? 

Reviewer #2: Yes

4. Have the authors made all data underlying the findings in their manuscript fully available?

Reviewer #2: Yes

5. Is the manuscript presented in an intelligible fashion and written in standard English?

Reviewer #2: Yes

6. Review Comments to the Author

Reviewer #2: I thank the authors for their additional experiments and revisions from the previous review. For the most part all comments were addressed except for the following:

The authors clearly demonstrate that leucine and KIC levels are elevated at 30 minutes and significantly decrease in most tissues by 60 minutes following gavage, yet the insulin tolerance tests performed occurred at 60 minutes. The authors argue based on these tolerance tests that there is no insulin resistance caused by leucine and KIC, yet their time point is clearly after levels have returned to normal physiological levels and no longer match levels cited in their discussion as being pathological.

The response provided to this comment does not adequately address the concern provided. As demonstrated by the data provided, levels of circulating leucine concentrations decrease at 60min from the peak at 30min, and in tissues the levels return to the same as control (water gavage). The argument that leucine levels remaining at 100 umole/liter is still quite high and thus similar to those seen with obesity and insulin resistance contradicts the findings in the works cited. Citation 45 shows that circulating leucine levels in nonobese subjects were 112uM and citation 46 shows that in normoglycemic that it was 113uM. In both citations levels were above 120uM if not above 130uM in groups with obesity and/or insulin resistance, suggesting that there is likely a minimum threshold to impact physiology, a threshold not being met at 60 minutes. This implication is supported by your findings as levels dramatically reduce at the tissue level when circulating levels decrease below this threshold. It is completely reasonable based on the veterinary recommendation that this limitation can not be addressed via experimentation, but this needs to be addressed in the discussion as the current study only suggests that the effects of leucine are transient and likely concentration dependent, and not the conclusion reached in lines 491-495.

7. PLOS authors have the option to publish the peer review history of their article (what does this mean?). If published, this will include your full peer review and any attached files.

Reviewer #2: No

---

## [Author Response · Author response to Decision Letter 1]

19 Jul 2024

We thank the associate editor and the reviewers for carefully reviewing the edits we made to our manuscript and the opportunity to fix this revision. We have thoroughly reviewed the manuscript to reflect the concern raised. In the response below, references to page and line numbers are for the clean copy of the revised manuscript.

Reviewer 2:

I thank the authors for their additional experiments and revisions from the previous review. For the most part all comments were addressed except for the following:

The authors clearly demonstrate that leucine and KIC levels are elevated at 30 minutes and significantly decrease in most tissues by 60 minutes following gavage, yet the insulin tolerance tests performed occurred at 60 minutes. The authors argue based on these tolerance tests that there is no insulin resistance caused by leucine and KIC, yet their time point is clearly after levels have returned to normal physiological levels and no longer match levels cited in their discussion as being pathological.

The response provided to this comment does not adequately address the concern provided. As demonstrated by the data provided, levels of circulating leucine concentrations decrease at 60min from the peak at 30min, and in tissues the levels return to the same as control (water gavage). The argument that leucine levels remaining at 100 umole/liter is still quite high and thus similar to those seen with obesity and insulin resistance contradicts the findings in the works cited. Citation 45 shows that circulating leucine levels in nonobese subjects were 112uM and citation 46 shows that in normoglycemic that it was 113uM. In both citations levels were above 120uM if not above 130uM in groups with obesity and/or insulin resistance, suggesting that there is likely a minimum threshold to impact physiology, a threshold not being met at 60 minutes. This implication is supported by your findings as levels dramatically reduce at the tissue level when circulating levels decrease below this threshold. It is completely reasonable based on the veterinary recommendation that this limitation can not be addressed via experimentation, but this needs to be addressed in the discussion as the current study only suggests that the effects of leucine are transient and likely concentration dependent, and not the conclusion reached in lines 491-495

Authors’ Response

We thank the reviewer for taking the time to read the revisions we made to the manuscript and for the reviewer’s comment. Although plasma leucine level was still significantly much higher at the time of the ITT (at least 3X the values in the water gavage group), we agree with the reviewer that there might be a threshold that is higher than what was observed at 60 min after gavage, especially in the tissues. Therefore, we have edited lines 491-498 (Discussion section) and lines 600-605 (Limitation section) to reflect this possibility. These sections now read:

Lines 491-498: “The fact that we observed no significant effect of leucine/KIC gavage on whole body insulin sensitivity in this study even with circulating levels of leucine and KIC that were much higher than basal (especially for leucine) suggests that altered leucine and KIC levels are a product of insulin resistance instead of causative. However, because the insulin tolerance test was carried out 60 min post gavage, at a time point when circulating and tissue levels of leucine and KIC were off their peak values (attained at 30 minutes), it is possible that the effects of leucine and KIC on insulin sensitivity are concentration dependent and transient.”

Lines 600-605: “Additionally, we performed the insulin tolerance test 60 min after gavage instead of at 30 min (when circulating and tissue leucine and KIC levels peaked). This decision was made based on the recommendation of the institutional animal care committee in order to reduce the stress of consecutive gavages/injections in a close time frame. By so doing however, we might have missed a critical time point because it is possible that the effect of the gavage on insulin sensitivity is concentration dependent and transient.”

---

## [Decision Letter · Decision Letter 2]

7 Aug 2024

KIC (ketoisocaproic acid) and leucine have divergent effects on tissue insulin signaling but not on whole-body insulin sensitivity in rats

PONE-D-24-07286R2

Dear Dr. Adegoke,

We’re pleased to inform you that your manuscript has been judged scientifically suitable for publication and will be formally accepted for publication once it meets all outstanding technical requirements.

Kind regards,

Owen Ngalamika

Academic Editor

PLOS ONE

Additional Editor Comments (optional):

Reviewers' comments:

Reviewer's Responses to Questions

**Comments to the Author**

1. If the authors have adequately addressed your comments raised in a previous round of review and you feel that this manuscript is now acceptable for publication, you may indicate that here to bypass the “Comments to the Author” section, enter your conflict of interest statement in the “Confidential to Editor” section, and submit your "Accept" recommendation.

Reviewer #2: All comments have been addressed

2. Is the manuscript technically sound, and do the data support the conclusions?

Reviewer #2: Yes

3. Has the statistical analysis been performed appropriately and rigorously? 

Reviewer #2: Yes

4. Have the authors made all data underlying the findings in their manuscript fully available?

Reviewer #2: Yes

5. Is the manuscript presented in an intelligible fashion and written in standard English?

Reviewer #2: Yes

6. Review Comments to the Author

Reviewer #2: (No Response)

7. PLOS authors have the option to publish the peer review history of their article (what does this mean?). If published, this will include your full peer review and any attached files.

Reviewer #2: No

---

## [Editor Report · Acceptance letter]

12 Aug 2024

PONE-D-24-07286R2 

PLOS ONE

Dear Dr. Adegoke, 

I'm pleased to inform you that your manuscript has been deemed suitable for publication in PLOS ONE. Congratulations! Your manuscript is now being handed over to our production team.

Kind regards, 

on behalf of

Dr. Owen Ngalamika 

Academic Editor

PLOS ONE